
# Developments in large-scale coastal flood hazard mapping
Michalis I. Vousdoukas[1,2], Evangelos Voukouvalas[1], Lorenzo Mentaschi[1], Francesco Dottori[1],
Alessio Giardino[3], Dimitrios Bouziotas[1,3], Alessandra Bianchi[1], Peter Salamon[1], Luc Feyen[1]
[1] [1] European Commission, Joint European Research Centre (JRC), Institute of Environment and Sustainability (IES),
Climate Risk Management Unit, Via Enrico Fermi 2749, I-21027-Ispra, Italy,
[2] Department of Marine Sciences, University of the Aegean, University hill, 41100, Mitilene, Lesbos, Greece
[3] Deltares, P.O. Box 177, 2600 MH Delft, The Netherlands.
*Correspondence to*: Michalis I. Vousdoukas (michalis.vousdoukas@jrc.ec.europa.eu)
**Abstract.** Coastal flooding related to marine extreme events has severe socio-economic impacts, and even though the
latter are projected to increase under the changing climate, there is a clear deficit of information and predictive capacity
related to coastal flood mapping. The present contribution reports on efforts towards a new methodology for mapping
coastal flood hazard at European scale, combining (i) the contribution of waves to the total water level; (ii) improved
inundation modelling; and (iii) an open, physics-based framework which can be constantly upgraded, whenever new
and more accurate data become available. Four inundation approaches of gradually increasing complexity and
computational costs were evaluated in terms of their applicability for large-scale coastal flooding mapping: static
inundation (SM); a semi-dynamic method, considering the water volume discharge over the dykes (VD); the Flood
Intensity Index approach (Iw); and the model LISFLOOD-FP (LFP). A validation test performed against observed
flood extents during the Xynthia storm event showed that SM and VD can lead to an overestimation of flood extents
by 232% and 209%, while Iw and LFP showed satisfactory predictive skill. Application at pan-European scale for the
present-day 100-year event confirmed that static approaches can overestimate flood extents by 56% compared to LFP;
however, Iw can deliver results of reasonable accuracy in cases when reduced computational costs are a priority.
Moreover, omitting the wave contribution in the extreme TWL can result in a ~60% underestimation of the flooded
area. The present findings have implications for impact assessment studies, since combination of the estimated
inundation maps with population exposure maps revealed differences in the estimated number of people affect within
the 20-70% range.
## 1    Introduction
During recent years our societies have witnessed several extreme meteorological events which have raised public
awareness to the fact that the climate is constantly changing and having a stronger footprint on everyday lives
compared to previous decades. Given that a large part of the world's population lives near the coast, the ongoing Sea
Level Rise (SLR) (DeConto and Pollard, 2016; IPCC, 2014) and its potential consequences have raised a lot of
attention; initially among the scientific community, but also from the side of stakeholders, governments and the public.
There is a great amount of recent studies which highlight that SLR will expose the coastal zone to greater risk in the
years to follow (Hinkel et al., 2014; Losada et al., 2013; Weisse et al., 2014); while several others project that more
frequent extreme weather events will enhance the impact of SLR on the coast (Brown et al., 2012; Debernard and





Røed, 2008; Gaslikova et al., 2013; Lowe et al., 2009; Vousdoukas et al., 2016). During extreme events the energetic
atmospheric conditions, result in transfer of mass and energy in the water element, which through the interaction with
the bathymetry are manifested as increased water levels. When the latter coincide with spring tides they can lead to
extreme events, affecting landward areas which normally are protected by water (Barnard et al., 2015; Bertin et al.,
39  2014).

The world's oceans are constantly exposing the coastal zone to energy fluxes, which are absorbed through dissipation
and sediment transport processes; driving the coastal morphology to states, which are the most effective in attenuating
ocean energy. During extreme conditions most hydro- and morpho-dynamic processes are accelerated, with the most
dramatic implication being the fact that the water level can exceed the height of natural (e.g. dunes, cliffs), or anthropic
barriers (e.g. seawalls, dykes) and reach areas not prepared to interact with the water element, with often catastrophic
consequences. This is the reason that marine storms are considered as extreme when they coincide with coastal
inundation, and inundation maps are a crucial element for several coastal management and engineering practices; i.e.
post-evaluation of extreme events, coastal planning, definition of set-back lines (Ferreira et al., 2006), and evaluation
of adaptation options (Cooper and Pile, 2014; Hinkel et al., 2010).
The static inundation approach ('bath-tub') considers as  flooded all the areas with elevation lower than the forcing
water level, comes with low computational costs, can be easily performed in GIS environments (Seenath et al., 2016)
and for that reason has been extensively used for studies of different scales (Hinkel et al., 2014; Hinkel et al., 2010;
Vousdoukas et al., 2012c). However, given the high complexity of coastal flooding processes, several recent studies
which showed that that the static approach results in substantial overestimation of the flood extent compared to
dedicated hydraulic models, especially in flatter terrains (Breilh et al., 2013; Gallien, 2016; Ramirez et al., 2016;
Seenath et al., 2016).
As intermediate solutions, approaches have been developed which are capable of reducing the computational cost by
taking into consideration either only water mass conservation (Breilh et al., 2013), or aspects of flooding
hydrodynamics (Dottori et al., 2016). A step more elaborate and more computationally intensive are dynamic, reduced
complexity models like LISFLOOD-FP (Bates et al., 2010), which despite being originally developed for simulating
river flow processes, have been proven to be reliable also for coastal flooding applications, such as the reproduction
of storm surge events (Ramirez et al., 2016; Smith et al., 2012) and the evaluation of future scenarios of sea level rise
(Purvis et al., 2008). At the same time, their application for large/continental (Alfieri et al., 2014) and global-scale
river flood mapping efforts (Sampson et al., 2015) is promising for their potential application also to coastal flooding,
but has not been explored yet. Finally, process-based models specialized for coastal hydro- and morpho-dynamics
(Lesser et al., 2004; McCall et al., 2010; Roelvink et al., 2009; Vousdoukas et al., 2012b) would appear as the optimal
option, however they come with the disadvantages of (i) the increased computational costs, which are almost
prohibitive for large scale application; and (ii) the fact that they require information about the nearshore topography
in detail which is often not available for many areas.
Despite the anticipated impacts of climate change along the world's coasts, there is a limited number of studies
evaluating the risk of coastal inundation along Europe or worldwide, while existing ones are based on the static
approach (Hinkel et al., 2014; Hinkel et al., 2010). Surprisingly, such large scale studies neglect the contribution of





waves to the extreme water levels, even though the latter has been shown to be important (Serafin and Ruggiero, 2014; Vousdoukas et al., 2012a). Against the foregoing background, the present study aims to propose a new methodology for mapping coastal flood hazard at European scale, by (i) considering the effect of waves when estimating extreme water levels; (ii) proposing the best method for coastal flood inundation mapping at continental scales, hereby trying to find a compromise between model complexity, data requirements *vs* availability, and constraints in computational power; and (iii) develop a framework which can be constantly upgraded every time new tools and data are available. To this end, four inundation approaches were tested and compared, initially on the grounds of their capacity to reproduce a historical extreme event. Following, the four approaches are applied and evaluated at European scale, on the grounds of the estimated flood extents, but also in combination with socio-economic information, in order to assess their effect on large-scale impact assessment of coastal flooding.

## 2 Data and methods

### 2.1 Total water level data

Extreme total water levels (TWL) are the result of the contributions from the mean sea level (MSL), the tide and the combined effect of waves and storm surge ($\eta_{\text{W-SS}}$):

$$TWL = \eta_{HTWL} + \eta_{W-SS}(t) \tag{1}$$

where $\eta_{\text{W-SS}}$ is becomes significant during extreme events, and $\eta_{\text{HTWL}}$ the high tide water level, defined as:

$$\eta_{HTWL} = MSL + \eta_{tide} \tag{2}$$

where $MSL$ is the Mean Sea Level, and $\eta_{\text{tide}}$ is the tidal elevation. The above values were estimated at ~11000 points, equally distributed every 25 km along the European coastline.

Time series of tidal elevation ($\eta_{\text{tide}}$) were obtained from the TOPEX/POSEIDON Global Inverse Solution (Egbert and Erofeeva, 2002), and 10 year data were analysed to obtain the maximum tide, given that the aim was extreme events. The TWL contribution due to extreme meteorological conditions $\eta_{\text{W-SS}}$ was reproduced by combining the effect of waves and storm surge:

- time series of extreme storm surge levels (*SSL*) were available from a storm surge hindcast run spanning from 01/01/1979 to 01/06/2014 (Vousdoukas et al., 2016). The simulations were carried out forcing the Delft3D-Flow module of the open source model Delft3D (Deltares, 2014) by atmospheric pressure and wind fields obtained from the ERA-Interim database (Dee et al., 2011). Detailed information can be found in Vousdoukas et al. (2016);
- time series of significant wave height $H_s$ were obtained by the ERA-INTERIM dataset (Dee et al., 2011).

The two datasets were combined to generate time series of the TWL component due to the combined effect of waves and storm surge according to the following equation:

$$\eta_{W-SS} = SSL + 0.2 \cdot H_s \tag{3}$$

where $0.2H_s$ is considered to be a reliable approximation of the wave setup; i.e. the elevation in mean water level near the coast due to wave shoaling and breaking (US Army Corps of Engineers, 2002). More elaborate ways to estimate





wave setup exist, considering apart from the significant wave height, also the wave period, length and beach slope.
However, information about the nearshore bathymetry and/or the slope is not available at European scale, at the
resolution required to resolve wave shoaling processes; therefore the solution was found to be the most reliable
approach.
Following, non-stationary extreme value statistical analysis (EVA) was applied to the 30-year $\eta_{W-SS}$ time series
allowing the estimation of extreme $\eta_{W-SS}$ values for different return periods. The statistical analysis consisted in (i)
transforming a non-stationary time series into a stationary one to which the stationary EVA theory can be applied; and
(ii) reverse-transforming the result into a non-stationary extreme value distribution and is described in detail in
Mentaschi et al. (2016). The values presently considered correspond to the 100-year present day event along Europe
(Figure 1).
The above imply that the pan-European application was simulating the hypothetical case that the 100-year event
occurred simultaneously along the entire European coastline. The increase in sea level during an extreme event is
episodic, and typically EVA provides only the TWL, and no information about the temporal evolution of the event.
This is a typical issue for similar studies and is usually dealt with the use of design hydrographs, such as the following
one (Cialone and Amein, 1993):
$$\eta_{W-SS}(t) = \eta_{peak}\left(1 - e^{\left|\frac{D}{t}\right|}\right)$$
(4)

where $\eta_{W-SS}(t)$ is the time varying water level above $\eta_{HTWL}$ due to the combined effect of waves and storm surge, $\eta_{peak}$
is the peak $\eta_{extreme}(t)$ value, $t$ the time and $D$ the half duration of the event. The event duration was considered to be a
function of $\eta_{peak}$ according to a linear relationship estimated for each point, estimated from the following procedure:
(i) the $\eta_{W-SS}$ water level time series was analyzed and extreme events were identified (in average 5 events per year);
(ii) for each event the duration and peak water level were estimated; (iii) a best-linear-fit relationship between $\eta_{peak}$
and $D$ was estimated for each point and was applied at the following stages of the analysis.
**2.2    Inundation modeling**
Four different inundation approaches were tested and are described from the most simplistic to the most elaborate and
computationally intensive one:
• Static inundation method in which areas hydraulically connected with the sea and below TWL are inundated

132       (SM);

• A semi-dynamic method, where the water volume discharge over the dykes is computed based on time series

134       of modelled water levels (VD); similar to the SO method described in Breihl et al (2013).

• The Flood Intensity Index approach (Iw) of Dottori et al. (2016). The index reproduces flooding processes

136       using an approximation of the water flow equations usually applied in two-dimensional hydraulic models,

137       considering the local topography, terrain roughness and basic information about the flood scenario.

• Dynamic inundation modeling using Lisflood-ACC (LFP) (Bates et al., 2010; Neal et al., 2011), a 2D

139       hydraulic model which is part of the Lisflood-FP model (Bates and De Roo, 2000). Lisflood-ACC has a one-



dimensional inertial model (e.g. advection is not considered) where *x* and *y* directions are decoupled in 2D
simulations over a raster grid. Recent work by Neal *et al*. (2011) showed that Lisflood-ACC is a faster
alternative to full shallow-water models for gradually varied subcritical flows; providing results of similar
accuracy as those of more complex models, both in terms of flow velocity and water depths, with a
considerably reduced computational effort.
Given that the spatial extent of the study area did not allow running simulations for the entire domain, the European
coastline was separated in ~11000 segments, each covering 25 km of shoreline and extending 100 km landward.
Elevation data for the flood simulations were taken from SRTM DTM at 3 arcseconds (~90m) resolution. For
simulations with the LFP and the Iw approach, hydraulic roughness values were derived from the CORINE Land
Cover map (Batista e Silva et al., 2012), as in Alfieri et al. (2014).
After the application of each approach the Flooded Area (FA) was estimated in km$^2$, while values were also aggregated
in country level, and normalized by country shoreline length; available from the World Resources Institute
([www.wri.org](http://www.wri.org)). In addition, FA values were grouped according to the geological characteristics of the coastline,
available from the European Environmental Agency ([www.eea.eu](http://www.eea.eu)). The dataset originally includes 20 geological
coastline classes; some of which were merged in order to reduce the total number to 12, with the mean FA estimated
for each shoreline class. Finally, the effect of the inundation approach on potential estimated number of people affected
by coastal flooding was assessed by combining the generated inundation maps with population maps at 100 m
resolution for Europe (Batista e Silva et al., 2013). The number of people affected was considered to be equal to the
total number of people located in areas predicted to be flooded.
**2.3    Integration of coastal protection structures**
Sufficient DEM resolution is crucial for inundation modelling and ideally <10 m resolution LIDAR data are
recommended for reliable results (Vousdoukas et al., 2012b; Vousdoukas et al., 2012c). However, such datasets are
often not available for continental scale studies; while such resolution implies computational costs which usually are
prohibitive. The 100 m resolution DEM presently used was a compromise between sufficient resolution and
computational effort, but was not sufficiently fine to resolve coastal protection structures, implying a potential
overestimation of inundation extents. Therefore, all available information on coastal protection structures in Europe
was compiled from open databases and national authorities (www.ahn.nl; UK Environmental Agency, pers. comm.;
Vafeidis et al., 2008).
The lack of detailed information about flood protection structures at European scale is a known issue (Scussolini et
al., 2015), and not all countries provide information with resolution fine enough for the analysis taking place in the
present study. Therefore, protection standards corresponding to the 5-year event were assumed along the areas for
which no data were provided, in order to avoid FA overestimation: the 5-year TWL was estimated from the extreme
value analysis and was considered as elevation of the coastal protection (Figure 2). Finally, the protection information
was introduced in the DEM by assigning the height of the coastal protection as elevation of all the DEM cells found
on the coastline and having elevation lower than the one of the protection (Figure 2).



### 2.4 Model validation

Model validation requires measurements from historical flooding events and in particular combination of water level time-series and flood extent maps for the same event. In general there is scarcity of well documented coastal inundation events; and according to our knowledge the Xynthia storm was the only large scale event which was sufficiently documented in Europe. Xynthia hit the Atlantic Coast of France in February 2010, causing the flooding of large coastal areas, with 47 deaths and at least 1.2 billion euros of damage (CGEDD, 2010). The coastal area located northward of the Gironde Estuary was the most severely affected, where flooded areas detected from satellites exceeded 300 km$^2$ and extensive information is available from reports and scientific literature (Bertin et al., 2012; Bertin et al., 2014; Breilh et al., 2013).

The coastline in the most flooded area is irregular and characterized by generally shallow sea floor area and large embayments, with extensive intertidal mudflats and coastal marshes. To prevent frequent marine flooding of these low-lying wetlands, an extensive system of dykes, levees and locks has been built over the last centuries, with an average height reported to be around 6 m. The elevation of the dykes was included in the DEM, however several dyke failures occurred during the event (Breilh et al., 2013) that have not been considered in the simulations, since their timing and location are unknown. Storm surge water levels were taken from observed water level at the La Pallice tide gauge, while flood extent was available from field measurements. River discharge has not been considered in the simulation, as the flooding event appeared to be mainly driven from high sea water levels; while river discharges were not significant.

The skill of the inundation approaches to reproduce the inundation events was evaluated on the grounds of agreement between simulated and observed flood footprints. Three different skill indexes were used, commonly applied for fluvial flooding (Alfieri et al., 2014; Bates and De Roo, 2000). The hit ratio $H$ is a proxy of agreement between simulated and observed inundation maps and it is defined as:

$$H = \frac{Fm \cap Fo}{Fo} \times 100 \tag{5}$$

where $Fm \cap Fo$ is the area correctly predicted as flooded by the model, and $Fo$ indicates the total observed flooded area. Since the hit ratio does not take into account over-prediction, the false alarm ratio $F$ was also considered, defined as:

$$F = \frac{Fm \, / \, Fo}{Fo} \times 100 \tag{6}$$

where $Fm/Fo$ is the area wrongly predicted as flooded by the model. Finally, a more comprehensive measure of the agreement between simulations and observations is given by the critical success index $C$, defined as:

$$C = \frac{Fm \cap Fo}{Fm \cup Fo} \times 100 \tag{7}$$

where $Fm \cup Fo$ is the union of observed and simulated flooded areas.





## 3 Results

### 3.1 Validation for the case of the Xynthia storm

The comparison of the observed inundation maps with the ones estimated by the different approaches, showed that the static approach largely overestimates the flood extent (Figure 3a), while taking into consideration the volume of water passing above the dykes (VD) improves marginally the performance (Figure 3b). Therefore, even though the hit rate for SM and VD was H>95%, F rates were higher than 200% and C rates around 25% (Figure 4). On the other hand the Iw and LFP approaches resulted in realistic flood extents, with the latter performing slightly better (Figure 3c-d). LFP resulted in higher Hit rates than Iw (73% and 84%, respectively), but also higher overestimation of the flood extents compared to Iw (F rates 47% and 68%, respectively). The two methods produced comparable results with C rate being slightly better for LFP compared to Iw (49% and 50%, respectively).

### 3.2 Coastal flooding hazard assessment at European scale

All four coastal inundation approaches were applied for the 11124 coastal segments along Europe in order to estimate flood extents (Figure 5), which were then aggregated at country level (Figure 6 and Table 1). The static approach resulted in the highest total FA (FA≈50381 km$^2$ for Europe), showing values substantially higher than the other approaches, especially along areas which are known for their low-lying/mild-slope terrains (e.g. North Sea; Figure 5a). The total flood extent for Europe based on VD was FA≈38613 km$^2$, slightly higher than the one for Iw (FA≈32510 km$^2$, Figure 5b-c), while LFP resulted in the lowest total flood extent overall, with FA≈30696 km$^2$ (Figure 5d). The spatial FA variations obtained from LFP and Iw were similar, in contrast to VD and SH, the results of which were characterized by some values which were substantially higher than the European mean.

### 3.3 Results per country and coastline type

Aggregating the FA values per country and normalizing per shoreline length showed that LFP and Iw resulted in relatively similar values (Figure 6c-d), with the exception of slightly higher Iw values for Germany and LFP values for Romania. The static approach resulted in higher FA per shoreline length, especially for Germany, Poland, UK and Italy (Figure 6a). Values from VD were overall varying within the ones of LFP and SM, with the exception of Germany for which the FA estimated from VD was the lowest among all approaches tested. Romania and Lithuania were the countries resulting in the higher FA per shoreline length in Europe.

Aggregating the FA values per coastline type showed that SM resulted in values higher than the other approaches by more than 30%, for all but three classes for which the differences were smaller: *Soft strands, Artificial beach* and *Small beaches* (Figure 7). Similarly to the previous findings, values from VD were higher than the ones of VD and LFP with the exception of three classes, for which VD produced the lowest values: *Artificial protection, Embankments* and *Muddy sediments*. Differences between LFP and Iw were small, with the former resulting in slightly higher FA for all classes apart from *Vegetative strands*.



### 3.4 Implications for coastal management and adaptation studies

Inundation maps are typically combined with socio-economic exposure maps to assess coastal impacts, or planning scenarios (Alfieri et al., 2015; Alfieri et al., 2016; Boettle et al., 2016; Prahl et al., 2015). Given that the number of people affected (NPA) is a parameter commonly considered and even used as a direct or indirect proxy of coastal impacts (Brown et al., 2013; Hinkel et al., 2010; Lloyd et al., 2015), the sensitivity of the estimated total NPA to the applied inundation approach was assessed. At this stage only SM and LFP were considered for reasons of simplicity; SM as the most common approach found in the literature, resulting in the higher flood extents (Figure 5a); and LFP being on the other extreme, producing the lowest FA values (Figure 5d) and being the most physically sound and complex approach to implement, among the ones tested.

SM resulted in 56% higher FA values than LFP for the whole of Europe, translated to a 65% increase in the NPA (~5 million instead of ~3; Figure 8). Not all countries showed the same sensitivity to the inundation approach used; e.g. relative differences in estimated FA from the two approaches reached, or even exceeded 50% for France, Italy, Romania, Portugal, Lithuania and the UK, but were <25% for the other countries. The above differences were also 'transferred' into NPA differences, but not in a linear way. The combination with the population maps resulted in higher NPA differences for Germany, Poland and Denmark, compared to the ones for FA; while relative NPA differences for France and Italy were reduced.

Including the wave contribution in the TWL estimation resulted in a ~150% increase in FA for the whole of Europe, with the relative FA differences exceeding 50%, with the exception of few countries like Estonia, Greece, Croatia, Lithuania, Romania and Turkey (Figure 9a). The increase in the European total NPA after including the wave effect was even higher, around 167% (~3.2 *vs* 1.2 million; Figure 9). The relative difference was higher than for FA for several countries, such as Germany, Denmark, Ireland, Latvia, Norway, and the UK (Figure 9b). Considering the wave effect was also shown to change the relative contribution of some countries to the European total, both for FA and NPA. For example UK, Norway, Germany and Denmark were shown to contribute more to the total once the waves were included in the analysis (Figure 9).

## 4 Discussion

### 4.1 Evaluation of inundation approaches

Validation of the static approach for the Xynthia storm showed that it results in severe overestimation of the flood extents in agreement with the findings of previous studies (Bertin et al., 2014; Gallien, 2016; Ramirez et al., 2016). The Iw and LFP approaches showed satisfactory predictive skill, which is an important finding since they were applied for Xynthia with the same setup as they were implemented for the entire European coastline, confirming the validity of the approach for large scale application.

Breihl et al. (2013) applied 3 different inundation approaches to simulate the Xynthia storm: (i) a static inundation approach forced by the maximum sea level recorded during the storm at La Pallice tide gauge (SM1); (ii) a second static approach which considers the space-varying maximum sea levels simulated by a storm surge modelling system (SM2); and (iii) the semi-dynamic VD method (VD-B2013). Their results are quantitatively similar even though they




cannot be directly comparable with the present ones, since Breihl et al. (2013) used a higher resolution DEM based
on LIDAR data, which can take into account for coastal defenses and sedimentary barriers, enhancing model
performance and allowing a more detailed analysis. Comparisons are more straightforward with results from Ramirez
et al. (2016) who obtained similar $C$ values running CEASAR-Lisflood on a STRM 90 m DEM, also concluding that
the static approach can overestimate FA by ~200%.
Dyke failure events were reported during Xynthia and since they were not taken into consideration in the simulations
they can be responsible for the weaker predictive skill in some areas. The latter could be partially compensated by
considering morphodynamic evolution during the inundation events, however such modeling is very computationally
expensive and thus not feasible at large scales; also due to the lack of essential data for such simulations (e.g. about
sediment characteristics). Overall, the results from the simulation of the Xynthia storm using Iw and LFP, show that
the latter can produce reliable results even when applied on a lower resolution DTM, which is an inevitable
compromise for large-scale applications, given the currently available computational power and data.

### 4.2   Towards an improved approach for pan-European coastal flood hazard mapping

The methodology for coastal inundation assessment presently proposed is improved in several aspects compared to
the current state of the art in large-scale coastal flood hazard mapping. Waves lead to an additional elevation in mean
water level near the coast due to wave shoaling and breaking, which during extreme events can be significant,
especially for exposed coastlines like the ones found along the Atlantic coast of Europe (Ciavola et al., 2011; Losada
et al., 2013; Serafin and Ruggiero, 2014). Nevertheless wave contribution is often neglected by existing large scale
studies and present results underline that omitting the wave effect can affect both the estimated FA and any consequent
impact calculations. The present efforts do not take into account all the wave-related processes contributing to coastal
flooding (i.e. erosion, overwash and breaching; e.g. Matias et al., 2008; McCall et al., 2010), as that would require
computationally intensive calculations and data which are not currently available on European scale. Still the approach
proposed is beyond the current state of the art and the differences in the estimated FA and NPA with and without
considering the wave contribution are significant.
Moreover, few studies exist which assess coastal inundation at European scale and overall, previous continental/global
scale efforts are based on the static inundation approach (Hinkel et al., 2014; Hinkel et al., 2010); which has been
shown to overestimate FA (see present findings; but also Bertin et al., 2014; Gallien, 2016; Ramirez et al., 2016). As
an improvement, the pan-European application shows that large-scale application of LFP is feasible, still the
computational effort implies the availability of a computational facility. When the latter is not available, Iw can be
considered as a valid alternative, as it was shown to produce comparable results with an order of magnitude lower
computational times.
The estimations of the number of people affected based on the produced inundation maps discussed in Section 3.4,
highlight that the increased complexity and computational effort related to the migration from the static to dynamic
inundation approaches can be outbalanced by the benefits in the quality of the produced results. High quality/detail
inundation maps are critical for coastal studies since the density of valuable assets often tends to increase landward
near the coast. The section stretching along the first hundreds of meters near the sea is acting as a buffer absorbing





energy from the ocean, and is typically too dynamic to host critical infrastructure. However, landward of that area the
density of population and valuable assets is typically high. Therefore overestimating flood extents is likely to result to
a disproportional increase in estimated impacts. The static approach was shown to result in overestimated flood extents
for coastline classes *Artificial protection, Harbor areas, Developed beaches,* and *Embankments,* which imply
increased socio-economic activity and high impact in case of flooding.
Ramirez et al. (2016) found that the static approach produced comparable results with CEASAR-Lisflood for the
Hurricane Sandy, a potential effect of the steep landscape. The latter was confirmed by the present findings reporting
smaller deviations between SH and the other approaches, for coastline classes typically associated with steep terrains
(i.e. *Cliffs*, *Artificial* and *Small beaches;* see Figure 7). On the contrary, higher deviations were observed for classes
associated with mildly sloping landscapes; i.e. *Estuary*, *Muddy sediments* and *Vegetative strands*.
A surprising finding in the comparison of the results from the pan-European application with the ones for the Xynthia
storm, was that while VD largely overestimated Xynthia FA, it produced results which were higher but comparable
to the ones from Iw and LFP for Europe. The reason could be that VD is sensitive to the protection standards
considered, as the height of the dykes controls the volume of water active in the inundation and consequently the flood
extent. The latter can be discerned by (i) the fact that estimated FA along the better protected North Sea coastline from
VD is lower than from Iw, while the opposite is the case for several less protected Mediterranean locations (Figure 5);
and (ii) that the VD produced the lowest FA values among all the methods for coastline class *Artificial protection* (see
Figure 7), which was typically not the case for other coastline types.
The quality of the information about coastal flood protection is critical for studies like the present one, and it is a
known issue that such data are not available along the entire European coastline at the resolution desired for the
inundation modelling. This has been also highlighted by previous studies on river flooding (Scussolini et al., 2015).
One potential solution would be to carry out reverse calculations of protection based on expected flood extents or
impacts, but this is still a challenge, given that such information is generally not available. At the present study it was
considered that the minimum protection standard applied in Europe was based on the 5-year event, which is probably
an overestimation of current flood protection, and implies that flood extents could be underestimated; especially at
some locations along the South European coastline. However, it is important to stress that the goal of the present
contribution is to establish a general framework for the assessment of flooding issues at the EU level, based on process-
based models and dynamic simulations. Both of these aspects are very novel in those type of studies. The proposed
framework allows for constant improvement of the quality of the results, whenever new and more accurate data will
become available.
**5    Conclusions**
A new methodology for mapping coastal flood hazard at European scale was presented, combining (i) the contribution
of waves to the total water level; (ii) improved inundation modelling; and (iii) an open, physics-based framework
which can be constantly upgraded, whenever new and more accurate data become available.
Four inundation approaches of gradually increasing complexity and computational costs were evaluated in terms of
their applicability for coastal flooding mapping along the European coastline: static inundation (SM); a semi-dynamic



method, considering the water volume discharge over the dykes (VD); the Flood Intensity Index approach (Iw); and
the model LISFLOOD-FP (LFP). To our knowledge, this is the first attempt to produce coastal flood hazard
estimations at continental scale using dynamic flood mapping approaches.
A validation test was performed against observed flood extents during the Xynthia storm event that occurred in 2010
in France. The results showed that SM and VD can lead to an overestimation of flood extents by 232% and 209%,
respectively; while Iw and LFP showed satisfactory predictive skill, especially considering that the setup for designed
for large-scale application, using a coarse 100 m DEM.
Application at pan-European scale for the present-day 100-year event confirmed that (i) static approaches can
overestimate flood extents by 56% compared to LFP; and that (ii) the latter can be applied successfully for large scale
studies. However, Iw can deliver results of reasonable accuracy in cases when reduced computational costs are a
priority.
The results showed that omitting the wave contribution in the extreme TWL can result in a ~60% underestimation of
the flooded area. Moreover, considering the wave contribution to the TWL changed the relative contribution of some
countries to the European total; due to the fact that for a part of the European coastline, waves are a more important
hazard component compared to storm surges.
The present findings have implications to impact assessment studies, since combination of the estimated inundation
maps with population exposure maps showed differences in the estimated number of people affect within the 20-70%
range.

## 6    Acknowledgments

The research leading to these results has received funding from the European Union Seventh Framework Programme
FP7/2007-2013 under grant agreement no 603864 (HELIX: "High-End cLimate Impacts and eXtremes";
www.helixclimate.eu), as well as by the JRC institutional projects Coastalrisk and GAP-PESETA II.

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




| | SM | VD | IW | LFP |
|---|---|---|---|---|
| **BELGIUM** | 0.0 | 0.0 | 0.0 | 0.0 |
| **BULGARIA** | 148.1 | 159.2 | 73.5 | 70.1 |
| **CYPRUS** | 100.3 | 117.6 | 92.9 | 69.5 |
| **GERMANY** | 5401.8 | 1485.8 | 3615.6 | 3051.0 |
| **DENMARK** | 4243.0 | 3077.4 | 3116.4 | 3201.1 |
| **ESTONIA** | 328.0 | 547.6 | 318.1 | 312.8 |
| **SPAIN** | 611.8 | 606.5 | 544.5 | 447.1 |
| **FINLAND** | 405.5 | 616.9 | 366.0 | 356.4 |
| **FRANCE** | 3202.6 | 996.9 | 1884.4 | 980.8 |
| **GREECE** | 2547.6 | 2877.0 | 2013.0 | 1924.7 |
| **CROATIA** | 621.1 | 1090.1 | 613.8 | 607.4 |
| **IRELAND** | 1712.9 | 2876.5 | 1590.9 | 1649.3 |
| **ITALY** | 5582.0 | 2428.3 | 2470.4 | 1916.3 |
| **LITHUANIA** | 1129.3 | 521.2 | 528.4 | 543.6 |
| **LATVIA** | 127.1 | 161.9 | 103.7 | 92.3 |
| **MALTA** | 10.9 | 15.7 | 10.9 | 7.2 |
| **NETHERLANDS** | 71.9 | 0.4 | 68.4 | 3.4 |
| **NORWAY** | 4936.6 | 8369.4 | 4967.5 | 4843.3 |
| **POLAND** | 1689.6 | 1252.7 | 782.3 | 861.9 |
| **PORTUGAL** | 343.8 | 200.9 | 251.1 | 171.0 |
| **ROMANIA** | 4408.7 | 2080.7 | 1314.5 | 1664.4 |
| **SWEDEN** | 1519.8 | 1989.3 | 1401.6 | 1269.1 |
| **SLOVENIA** | 24.9 | 44.6 | 21.9 | 23.2 |
| **TURKEY** | 1375.0 | 1725.0 | 868.9 | 877.8 |
| **UNITED KINGDOM** | 9910.5 | 5371.5 | 5491.8 | 5752.9 |
| **EU28** | 44141.1 | 28518.6 | 26674.2 | 24975.5 |
| **EU-TOTAL** | 50452.6 | 38613.0 | 32510.6 | 30696.6 |


**Table 1. Values of Flooded Area per EU country for the present day 100-year event (in km$^2$), obtained from the four tested**
**inundation approaches. Totals for EU and EU28 are also provided.**




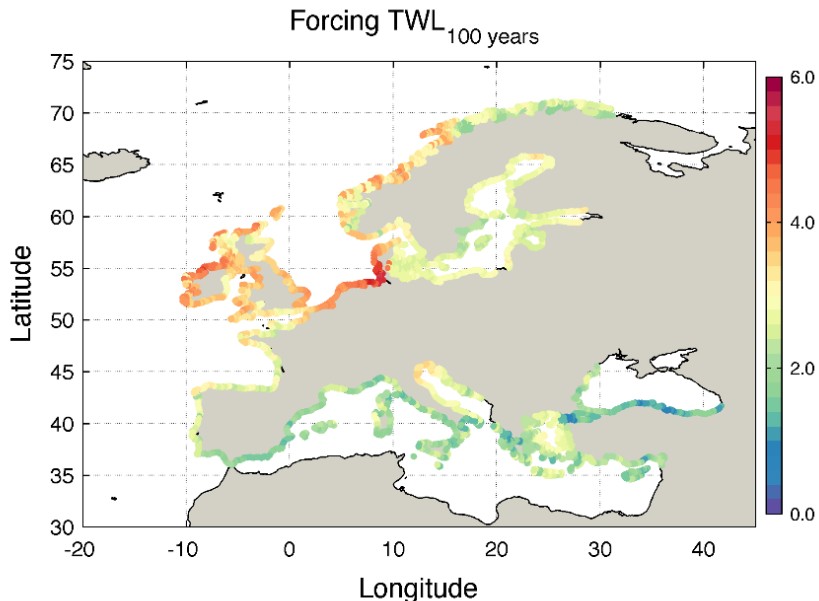


**Figure 1. Total Water Level values for the present day 100-year event along Europe; values are shown every 25 km of**
**coastline.**






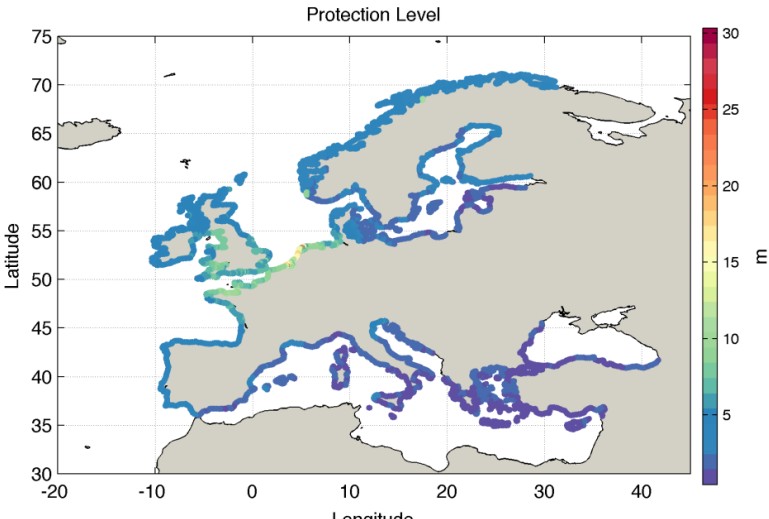


**Figure 2. Protection standards considered along the European coastline, expressed as Design Total Water Levels (TWL).**






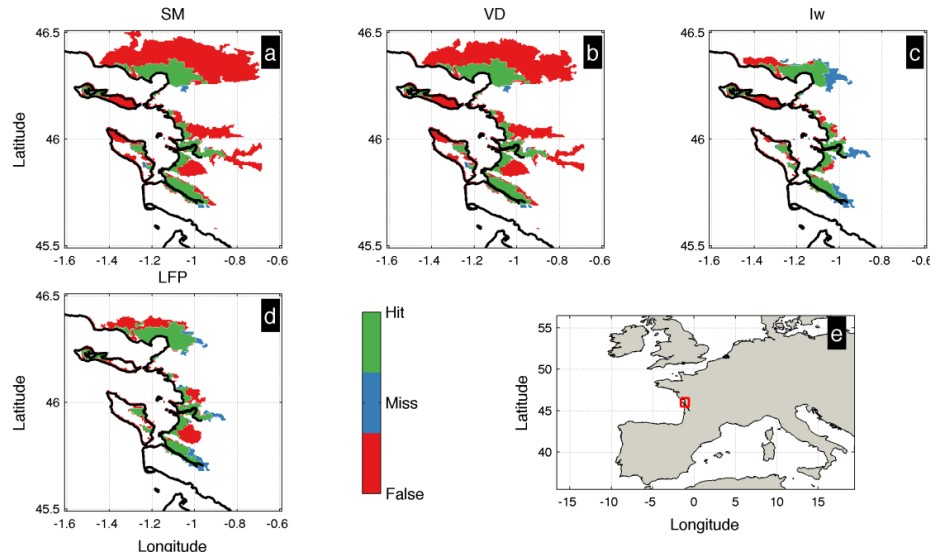


**Figure 3. Validation of the LISFLOOD-FP model for the Xynthia storm: Maps showing the comparison of the simulated and observed flood extent, as well as their intersection: green, blue and red colors correspond to inundated areas predicted, not- predicted and overpredicted by the model, respectively. Map (e) shows the location of the study area.**






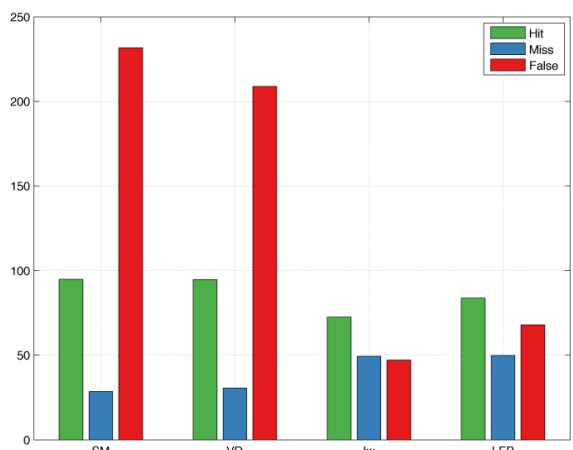


**Figure 4. Validation of the different inundation approaches (bar bundles) for the Xynthia storm, on the grounds of the H,**
**F and C rates (shown by different colors).**



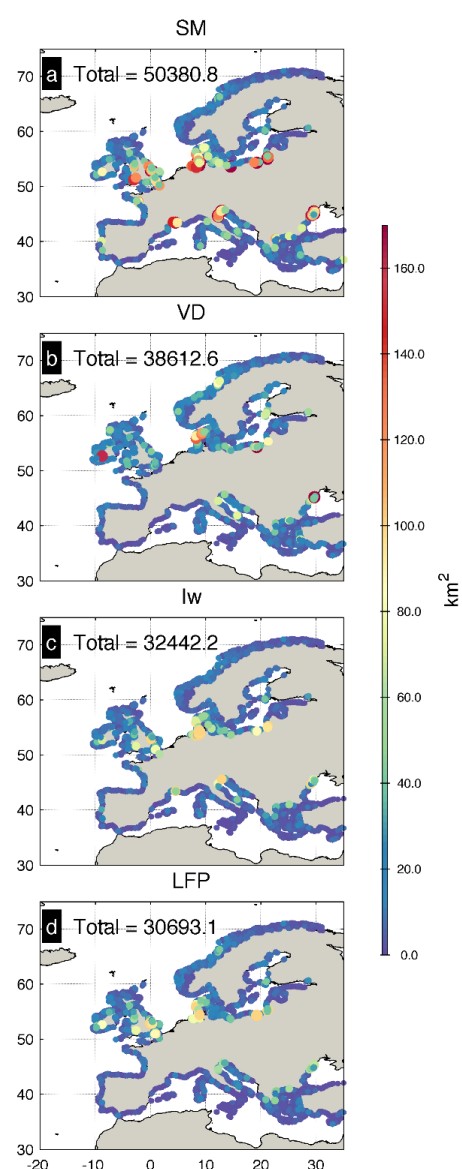


**Figure 5. Estimated coastal flood extent for the present day 100-year event using all four approaches. Values as shown for each 25 km coastal segment and correspond to km².**



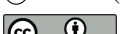


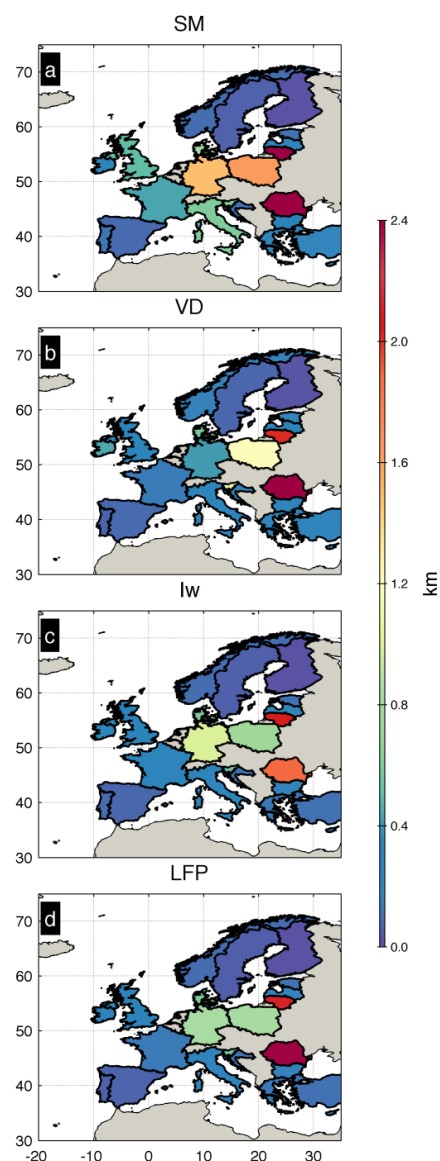


**Figure 6. Estimated coastal flood extent for the present day 100-year event using all four approaches, aggregated per country-level, and normalized by coastline length. Values correspond to km² per km of coastline.**







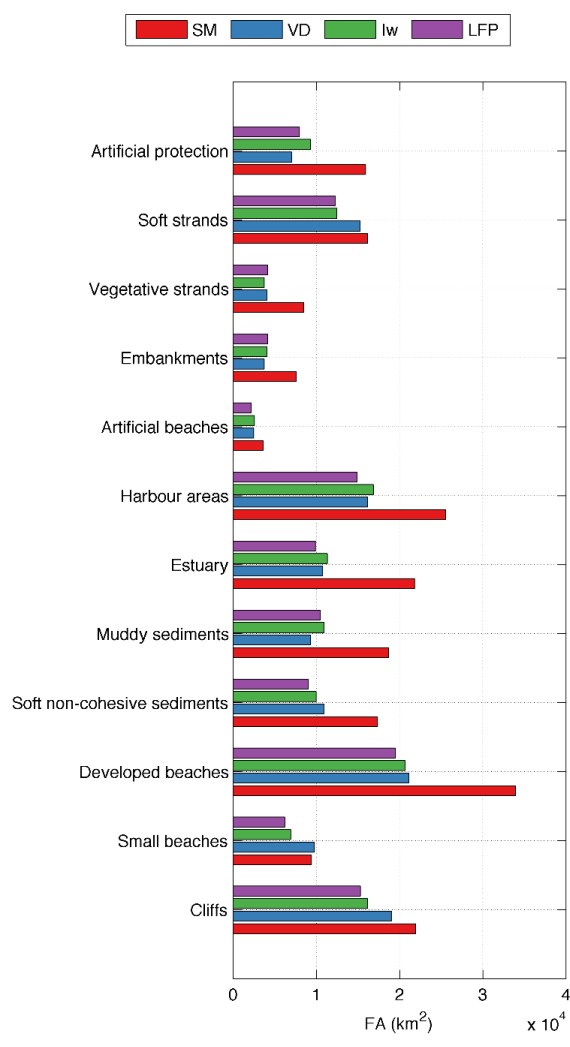


**Figure 7. Comparison of the flooded area (FA) for the present day 100-year event aggregated per coastline type for all four**
**inundation approaches. Values correspond to km² per km, colors express the different inundation approaches (see legend)**
**and horizontal bar stacks the shoreline type.**





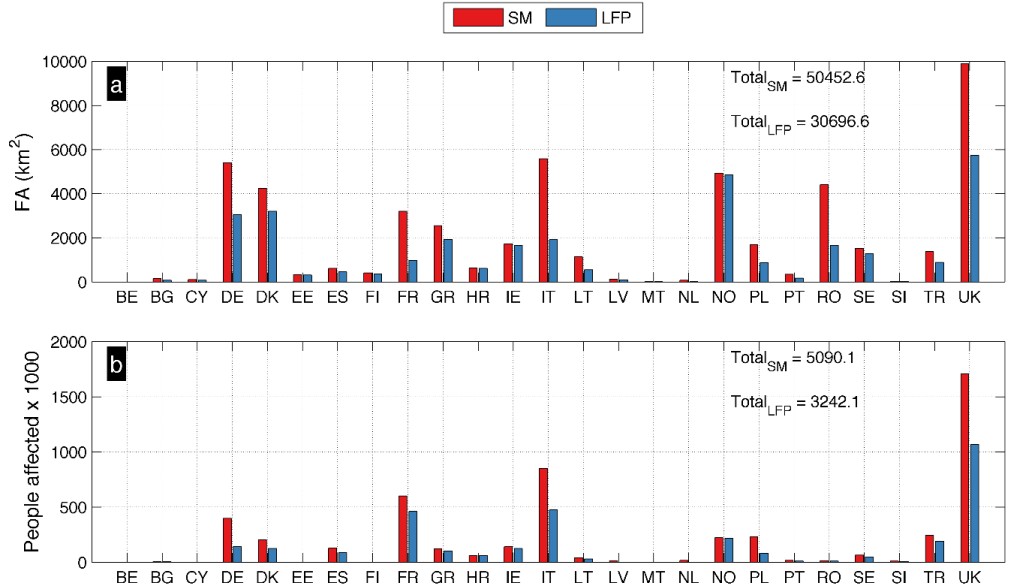


**Figure 8. Estimated values of the country level FA (a) and thousands of people affected (b) for the present-day 100-year event; comparisons between the results using the static approach and LISFLOOD-FP.**








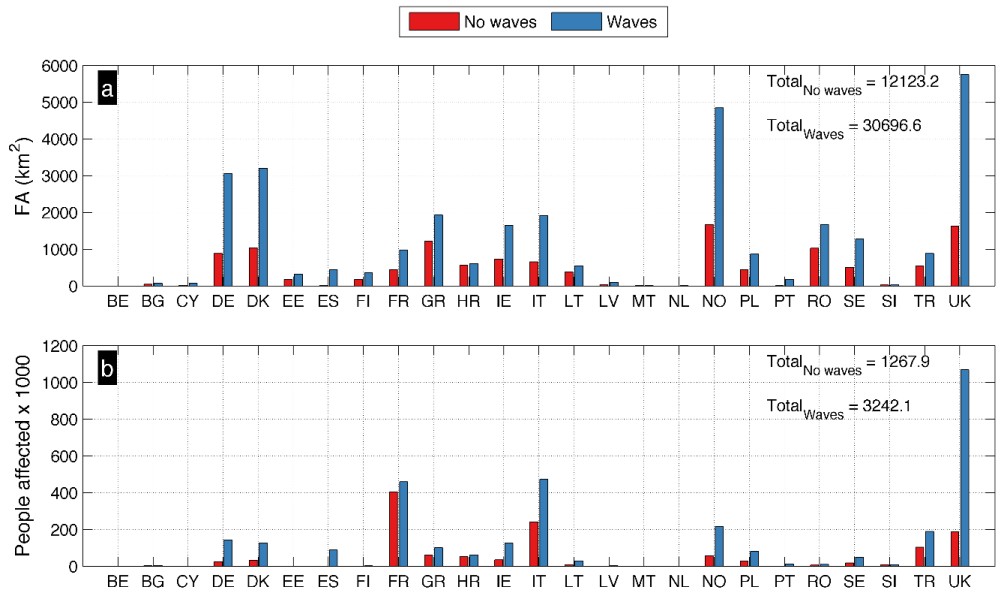

**Figure 9. Estimated values of the country level FA (a) and thousands of people affected (b) for the present-day 100-year event; comparisons between the results considering TWL including waves or not.**