# Peer review of "Developments in large-scale coastal flood hazard mapping 1"

_Natural Hazards and Earth System Sciences, 2016_

## Referee Comment (RC1) · Anonymous Referee #1 · 11 Jul 2016

The manuscript represents the very first approach to the problem of coastal flooding at an European scale. The paper is very well written and is based on a rigorous approach.

It is notable the fact that four different methodologies are compared, spacing from a simple bath-tub approach to more sophisticated numerical modeling.

Regarding the literature review a littlebit extra can be done. The author on page 2-lines 52-58 presents basically two approaches with different potential of application but also variable computing requirements, e.g. bath-tub in a GIS environment against numerical modeling. A third intermediate solution, a dynamic reduced model like LISFLOOD is finally considered.

In reality a fourth approach can be considered, the use of sophisticated GIS methodologies, e.g. the Cost-Distance Approach, thus introducing a "proxy" for inundated

surface characteristics, like the presence of obstacles or passageways. This new approach has been recently compared to more detailed hydrological estimations using wave run-up calculations and the results are promising. Likewise other approaches, it has been used for the application of the flood directory, the method and reference case study can be found in the recent paper on NHESS by Perini et al., 2016 (DOI: 10.5194/nhess-16-181-2016). Another interesting application is the one by Sekovski et al., 2015 (DOI: 10.5194/nhess-15-2331-2015), to predict increased vulnerability to urban growth.

Regarding the approximation that the authors used for the introduction of coastal structures in the DTM, described at page 5, lines 170-172, I think it is a clever solution. However, can they clarify why they use a 5 year return period level? Is not a littlebit low? My experience is that defense structures are built with an elevation which corresponds to much higher safety levels, if not, low-lying areas behind dykes would be flooded too regularly. Personally I would have a chosen at least a 10 return period level.

At page 6 the authors say that Xynthia is the best documented flood in Europe. This is not true, the 1953 flood in the Netherlands, Belgium and the UK is very well documented. If they mean that Xynthia is more documented regarding the flood extent using for example satellite-based methods I agree. But I believe the statement should be rephrased.

On the same page, at line 190 they say that the flood extend was obtained by field measurements. Who did these measurements? Are they publicly available? Also, the sentence at line 191 is a broken one.

Some reviewing is needed for the reference list.

1) The reference by Breilh et al. should come before Boetle et al. 2) Is the paper by Dottori already published? Add page numbers. 3) The paper by Vousdoukas et al. at line 483 is now published. Please upgrade.

---

## Referee Comment (RC2) · Anonymous Referee #2 · 19 Jul 2016

The paper describes the development of a coastal flooding methodology that is then applied at a European Scale. Undertaking a European-wide coastal flood mapping exercise is a complex and challenging task, that is not to be underestimated. There are well known data gathering and computational challenges that arise when undertaking studies at this scale. The authors are to be congratulated for their efforts and achievement.

As with all studies of this type it is inevitable there are significant uncertainties associated with the methodology and results. Presumably the main objective of the analysis, and perhaps this could be made clearer, is to enable the relative comparison of coastal flood risk for different regions in Europe? Hence, care should be taken when interpreting the results, particularly at local scales. Given the necessary methodological limitations it is perhaps worth expanding on those limitations within the text, as discussed further here.

The approach to extreme value modelling that has been adopted involves the application of what Bruun and Tawn (1998) termed the Structure Variable Method (SVM). The SVM involves the reduction of the multivariate sea condition to a univariate distribution, of set-up in this context, thus enabling univariate extreme value methods to be applied. There are a number of known limitations associated with this approach, Bruun and Tawn (1998).

In areas where the tidal regime is significant, the coastal flood response is sensitive to the timing of peak wave conditions. Peak wave conditions occurring at low tide versus high tide can mean the difference between severe or no flooding. The SVM implicitly assumes the distribution of the timing of peak wave conditions, in relation to the astronomical tide, is explicitly defined within the historical observations. Or, in other words, the SVM does not explicitly consider the likelihood that severe storms that, by chance, peaked (in terms of wave height) at low tide, could occur at high tide. This can lead to an underestimation in the extremes. The other main limitation of the SVM is extrapolation in the region where the variable itself (set-up in this case) maybe highly non-linear. The process of extrapolation will not capture these non-linearities and hence joint probability methods are often employed instead, Bruun and Tawn (1998), Hawkes et al (2002), Wahl et al (2012) and Gouldby et al (2014), for example.

The use of wave-setup as the variable for defining the peak sea condition level is also of interest. Coastal flooding can occur through processes of wave runup and associated wave overtopping. i.e. when the dynamic water level far exceeds the still water level (including setup). So whilst the wave effects have been included in this analysis, this is only a partial inclusion that does not include the dynamic wave proceeses. It would have been possible to utilise a wave runup formula, that includes the important variable of wave period but not necessarily beach slope (for which it is understood there are data restrictions) , Stockdon et al (2006), for example, to capture the dynamic wave effects. It would be interesting to understand the rationale for the alternative that was adopted
and perhaps extend the text to include this discussion.

Data limitations at this scale are well-known and the authors have overcome limitations relating to defence crest level data using a standard of protection (SOP) based approach that has been widely applied on previous studies. The choice of the 5-year SOP for areas where no defence information is available warrants further discussion. Where defences have been constructed these will often have been designed to have a standard of protection greater than 100 years. Would the methodology not therefore significantly overestimate flooding in these areas?

References

Bruun, J.T. and Tawn, J.A., 1998. Comparison of approaches for estimating the probability of coastal flooding. Journal of the Royal Statistical Society: Series C (Applied Statistics), 47(3): 405-423.

Gouldby B, Mendez F, Guanche Y, Rueda A and Minguez R (2014) A methodology for deriving extreme nearshore sea conditions for structural design and flood risk analysis, Coast. Eng, vol. 88, Pages 15-26, June

Hawkes, P.J., Gouldby, B.P., Tawn, J.A. and Owen, M.W., 2002. The joint probability of waves and water levels in coastal engineering design. Journal of Hydraulic Research, 40(3): 241-251

Stockdon, H.F., Holman, R.A., Howd, P.A. and Sallenger Jr, A.H., 2006. Empirical parameterization of setup, swash, and runup. Coastal Engineering, 53(7): 573-588.

Wahl, T., Mudersbach, C. and Jensen, J., 2012. Assessing the hydrodynamic boundary conditions for risk analyses in coastal areas: A multivariate statistical approach based on copula functions. Nat. Hazards Earth Syst. Sci., 12(2): 495–510

---

## Author Comment (AC1) · 21 Jul 2016

The manuscript represents the very first approach to the problem of coastal flooding at an European scale. The paper is very well written and is based on a rigorous approach. It is notable the fact that four different methodologies are compared, spacing from a simple bath-tub approach to more sophisticated numerical modeling. Regarding the literature review a little bit extra can be done. The author on page 2-lines 52-58 presents basically two approaches with different potential of application but also variable computing requirements, e.g. bath-tub in a GIS environment against numerical modeling. A third intermediate solution, a dynamic reduced model like LISFLOOD is finally considered. In reality a fourth approach can be considered, the use of sophisticated GIS methodologies, e.g. the Cost-Distance Approach, thus introducing a "proxy" for inundated surface characteristics, like the presence of obstacles or passageways.

[Figure]

This new approach has been recently compared to more detailed hydrological estimations using wave run-up calculations and the results are promising. Likewise other approaches, it has been used for the application of the flood directory, the method and reference case study can be found in the recent paper on NHESS by Perini et al., 2016 (DOI:10.5194/nhess-16-181-2016). Another interesting application is the one by Sekovski et al., 2015 (DOI: 10.5194/nhess-15-2331-2015), to predict increased vulnerability to urban growth.

Authors: We are thankful to the reviewer for the positive comments on our work. We agree that the introduction could be improved and in the revised version we are discussing the proposed methodologies, as well as others proposed by the second referee.

Regarding the approximation that the authors used for the introduction of coastal structures in the DTM, described at page 5, lines 170-172, I think it is a clever solution. However, can they clarify why they use a 5 year return period level? Is not a little bit low? My experience is that defense structures are built with an elevation which corresponds to much higher safety levels, if not, low-lying areas behind dykes would be flooded too regularly. Personally I would have a chosen at least a 10 return period level.

Authors: The accuracy, detail, and spatial resolution/density of the available information about coastal protection varies substantially among countries; i.e. for some countries detailed GIS layers are available, while for others little information can be found. This is a known shortcoming, which has been acknowledged also by both reviewers. As a result collecting and improving the information on coastal protection has been a constant task during the last years. Along most urban centers it was possible to have at least a rough estimate, i.e. from personal communication with national authorities or the coastal engineering community. The same applies for some countries (e.g. Belgium) for which data were not officially available. Therefore most urban centers have been considered to be protected by more rare events and the 5-year event standard has been applied mainly along areas for which we had no information. Those were

mostly locations with low population density. There, a protection standard even lower than for the 5-year event is often in place and for that reason we expect that our results could be even conservative.

At page 6 the authors say that Xynthia is the best documented flood in Europe. This is not true, the 1953 flood in the Netherlands, Belgium and the UK is very well documented. If they mean that Xynthia is more documented regarding the flood extent using for example satellite-based methods I agree. But I believe the statement should be rephrased.

Authors: The comment is fair and the statement has been rephrased. Xynthia is not the only documented event, but most likely the only recent well documented event.

On the same page, at line 190 they say that the flood extend was obtained by field measurements. Who did these measurements? Are they publicly available?

Authors: The flood extent information was obtained by digitizing reports and papers, and references have been added in the revised manuscript (Breilh et al. 2013 and DDTM-17, 2011). Similar information could be also found in the following website: http://www.storm-surge.info/data-access, even though they were not used in the present work.

Also, the sentence at line 191 is a broken one.

Authors: The sentence has been rephrased.

Some reviewing is needed for the reference list. 1) The reference by Breilh et al. should come before Boetle et al. 2) Is the paper by Dottori already published? Add page numbers. 3) The paper by Vousdoukas et al. at line 483 is now published. Please upgrade.

Authors: The reference list has been corrected. Vousdoukas et al and Dottori et al are still online as corrected proofs. The order of the references is made by the Endnote reference manager software using the journal template and should be correct. It will

be further checked and corrected in case of a problem before final publication.

---

## Author Comment (AC2) · 21 Jul 2016

The paper describes the development of a coastal flooding methodology that is then applied at a European Scale. Undertaking a European-wide coastal flood mapping exercise is a complex and challenging task, that is not to be underestimated. There are well known data gathering and computational challenges that arise when undertaking studies at this scale. The authors are to be congratulated for their efforts and achievement.

Authors: We are thankful to the reviewer for the positive comments.

As with all studies of this type it is inevitable there are significant uncertainties associated with the methodology and results. Presumably the main objective of the analysis, and perhaps this could be made clearer, is to enable the relative comparison of coastal

flood risk for different regions in Europe? Hence, care should be taken when interpreting the results, particularly at local scales. Given the necessary methodological limitations it is perhaps worth expanding on those limitations within the text, as discussed further here. The approach to extreme value modelling that has been adopted involves the application of what Bruun and Tawn (1998) termed the Structure Variable Method (SVM). The SVM involves the reduction of the multivariate sea condition to a univariate distribution, of set-up in this context, thus enabling univariate extreme value methods to be applied. There are a number of known limitations associated with this approach, Bruun and Tawn (1998). In areas where the tidal regime is significant, the coastal flood response is sensitive to the timing of peak wave conditions. Peak wave conditions occurring at low tide versus high tide can mean the difference between severe or no flooding. The SVM implicitly assumes the distribution of the timing of peak wave conditions, in relation to the astronomical tide, is explicitly defined within the historical observations. Or, in other words, the SVM does not explicitly consider the likelihood that severe storms that, by chance, peaked (in terms of wave height) at low tide, could occur at high tide. This can lead to an underestimation in the extremes. The other main limitation of the SVM is extrapolation in the region where the variable itself (set-up in this case) maybe highly non-linear. The process of extrapolation will not capture these non-linearities and hence joint probability methods are often employed instead, Bruun and Tawn (1998), Hawkes et al (2002), Wahl et al (2012) and Gouldby et al (2014), for example.

Authors: We fully agree with the reviewer that a properly implemented multivariate approach would be more appropriate and all this is discussed in the revised manuscript (line 342). As mentioned in the revision, one of the main future aims is to assess impacts from coastal flooding in view of climate change and for that reason the emphasis was put in developing a non-stationary statistical approach. Ongoing efforts include a sensitivity analysis of the uncertainty from different parameters and some preliminary results can be found in the following MSc thesis: http://repository.tudelft.nl/islandora/object/uuid%3A06e553ce-f491-4bf3-badd-

58978f4fe7ac?collection=education. Moreover, we are working on a non-stationary, multivariate approach, but this is not a trivial task given all the different variables involved.

The use of wave-setup as the variable for defining the peak sea condition level is also of interest. Coastal flooding can occur through processes of wave runup and associated wave overtopping. i.e. when the dynamic water level far exceeds the still water level (including setup). So whilst the wave effects have been included in this analysis, this is only a partial inclusion that does not include the dynamic wave proceeses. It would have been possible to utilise a wave runup formula, that includes the important variable of wave period but not necessarily beach slope (for which it is understood there are data restrictions) , Stockdon et al (2006), for example, to capture the dynamic wave effects. It would be interesting to understand the rationale for the alternative that was adopted and perhaps extend the text to include this discussion.

Authors: We agree that wave run up is important and we were tempted to include it in the analysis, given also the fact that some of the authors have extensive experience on the topic. We have added a relevant paragraph in the discussion section (line 335) and the main reasons for not including wave run up are: - Run up elevation is directly related to the topography for which we don't have information (as also mentioned by the reviewer), i.e. dissipative beaches usually come with higher set-up and lower run-up heights (more energy lost during shoaling/breaking), while the contrary applies to reflective beaches. Also not all run up is not relevant for the entire EU coastline, a big part of which consists of rocks/cliffs. For that reason we found the generic approximation of wave setup as a suitable generic solution. - swash is a higher frequency process and the present inundation modeling is focusing on the rare extreme events, which drive a slow and persistent increase in sea levels. We were not confident about adding the R2% run up height to the tidal and storm surge elevation, since swash elevation is fluctuating around the wave setup elevation and R2% is reached only occasionally during the event. Moreover the period of the swash fluctuations is related to the wave

frequencies attenuated (or not) along the surf zone, again related to whether the beach tends to be dissipative or reflective. All the above would make the correct incorporation of wave run-up a challenge and for that reason using only wave setup was chosen as a more sound approach.

Data limitations at this scale are well-known and the authors have overcome limitations relating to defence crest level data using a standard of protection (SOP) based approach that has been widely applied on previous studies. The choice of the 5-year SOP for areas where no defence information is available warrants further discussion. Where defences have been constructed these will often have been designed to have a standard of protection greater than 100 years. Would the methodology not therefore significantly overestimate flooding in these areas?

Authors: There is a similar comment from referee 1 and the reply is added below: "The accuracy, detail, and spatial resolution/density of the available information about coastal protection varies substantially among countries; i.e. for some countries detailed GIS layers are available, while for others little information can be found. This is a known shortcoming, which has been acknowledged also by the reviewers. As a result collecting and improving the information on coastal protection has been a constant task during the last years. Along for most urban centers it was possible to have at least a rough estimate, i.e. from personal communication with national authorities or the coastal engineering community. The same applies for some countries (e.g. Belgium) for which data were not officially available. Therefore most urban centers have been considered to be protected by more rare events and the 5-year event has been considered only at areas along which we had no information and those were mostly locations with low population density. There, a protection standard even lower than for the 5-year event is often in place and for that reason we expect that our results could be even conservative."